# Stem Cell Therapy for Post-Traumatic Stress Disorder: A Novel Therapeutic Approach

**DOI:** 10.3390/diseases9040077

**Published:** 2021-10-29

**Authors:** Dhir Gala, Vikram Gurusamy, Krishna Patel, Sreedevi Damodar, Girish Swaminath, Gautam Ullal

**Affiliations:** Department of Neuroscience, American University of the Caribbean School of Medicine, 1 University Drive at Jordan Dr, Cupecoy, Sint Maarten; vikramgurusamy@students.aucmed.edu (V.G.); krishnakpatel@students.aucmed.edu (K.P.); sreedevidamodar@students.aucmed.edu (S.D.); girishswaminath@students.aucmed.edu (G.S.); gullal@aucmed.edu (G.U.)

**Keywords:** stem cell therapy, pluripotent stem cells, cell therapy, applications, post-traumatic stress disorder, epilepsy

## Abstract

Stem cell therapy is a rapidly evolving field of regenerative medicine being employed for the management of various central nervous system disorders. The ability to self-renew, differentiate into specialized cells, and integrate into neuronal networks has positioned stem cells as an ideal mechanism for the treatment of epilepsy. Epilepsy is characterized by repetitive seizures caused by imbalance in the GABA and glutamate neurotransmission following neuronal damage. Stem cells provide benefit by reducing the glutamate excitotoxicity and strengthening the GABAergic inter-neuron connections. Similar to the abnormal neuroanatomic location in epilepsy, post-traumatic stress disorder (PTSD) is caused by hyperarousal in the amygdala and decreased activity of the hippocampus and medial prefrontal cortex. Thus, stem cells could be used to modulate neuronal interconnectivity. In this review, we provide a rationale for the use of stem cell therapy in the treatment of PTSD.

## 1. Introduction

Post-traumatic stress disorder (PTSD) is a psychiatric disorder characterized by difficulty recovering from an exposure to an exceptionally threatening/horrifying event or to a prolonged trauma. Exposures may include serious accidents, natural disasters, combat/war, rape or death threats [1]. PTSD presents as persistence of intrusive recollections, avoidance of trauma-related stimuli, negative alterations in mood, and hyperarousal symptoms for greater than one month [1]. Events can be relived through nightmares and flashbacks leading to feelings of sadness, anger, or fear and further leading to detachment from others. It can present as a significant health burden by increasing the risk of suicide and other medical conditions [2]. The lifetime prevalence of PTSD is between 1.5% and 8.8%. The prevalence among U.S. adults is 3.5%, with women twice as likely to have PTSD than men. PTSD is highest among U.S. African Americans, Native Americans, and Hispanics/Latinos when compared to non-Hispanic/Latino Caucasians [3,4,5].

Treatment options for PTSD include psychotherapy and medication, either used alone or in combination. Cognitive processing therapy addresses the negative thoughts surrounding the trauma and confront distorted thought patterns in order to change how they feel or act. Prolonged exposure therapy uses detailed repetitive recollection of the trauma to expose the patient to their symptoms in a controlled and safe environment to improve coping skills [6]. Group therapy allows survivors of traumatic events to feel comfortable and supported in sharing their emotions in a non-judgmental setting. Other psychotherapies include interpersonal, supportive, and psychodynamic therapies focusing on self-regulation and interpersonal relationships [6]. Medications can be used in PTSD for symptomatic relief; however, their efficacy and response rates are low with Selective serotonin reuptake inhibitors (SSRIs) such as fluoxetine and paroxetine showing limited efficacy over placebos [7]. Other medications such as serotonin-norepinephrine reuptake inhibitors (SNRIs) have also shown minimal benefit for PTSD symptoms. The first-line medications used for PTSD include sertraline, fluoxetine, paroxetine, and venlafaxine. Prazosin is an alpha-1 adrenergic antagonist used to reduce nightmares in PTSD, but studies show variability in its efficacy for symptom improvement. The prevalence of treatment-resistant PTSD is approximately 33%. Current non-pharmacological therapies for PTSD have limited availability and pharmacological management can lead to significant adverse effects [8]. PTSD is a chronic condition for many patients and commonly recurs or resists treatment, suggesting the need to explore further therapeutic options.

The pathophysiology of PTSD involves abnormal neural connectivity between the amygdala, hippocampus, and medial prefrontal cortex (mPFC). PTSD and temporal lobe epilepsy have similar abnormal neuroanatomic neural networks. Both disorders have functional abnormalities in the anterior mesial temporal lobe and neural pathways involving the temporal lobe implicating similar therapeutic options could be used for both disorders [9,10,11,12,13]. In epilepsy, repeated abnormal paroxysmal electrical discharges and seizure activities lead to progressive neuronal cell damage [14]. In temporal lope epilepsy, the hippocampus becomes the common target of neuronal injury. This can induce neurogenesis, apoptosis, and neuronal impairment with limited GABAergic receptor expression resulting in anxiety, depression, and cognitive impairment [15]. More recently, epilepsy has been considered a candidate for stem cell therapy for use in prophylaxis and enhancement of cognitive function. By using stem cell therapy and modifying the expression of specific neurotrophic factors, impairment can be reduced in the affected areas and provide better outcomes [16]. Studies in children have shown use for stem cell therapy in treatment-resistant cases [17]. Targeting stem cell therapy to the amygdala and hippocampal regions of the brains can further help to manage the complications of epilepsy [18]. Stem cell therapy has been used to treat the neuropsychiatric components of epilepsy. By nature of their similarities in neuroanatomic neural networks, stem cells could be utilized for the treatment of PTSD. Loss or over-activation of neuronal nuclei or their projections in the amygdalohippocampal circuit are implicated in the pathogenic mechanism of both epilepsy and PTSD. Therefore, we suggest that stem cell therapy targeting specific regions of the brain to enhance the growth of new neurons could be used in the treatment of PTSD, to reduce symptoms and potentially address the underlying cause of PTSD.

## 2. Stem Cell Therapy: Terminology and Cell Lines

### 2.1. Overview

Stem cell therapies are being extensively used in regenerative medicine, especially in neurological pathologies, due to the low regenerative potential of the central nervous system (CNS). Numerous pre-clinical animal models have shown stem cell therapy to be safe and effective leading to an increasing number of clinical trials [19]. This surge in clinical trials has led to the formation of regulations and guidelines for the use of stem cells in treating various neurological disorders [20,21]. Stem cell therapies provide benefit through various mechanisms in the CNS such as replacement of cells, modulating the inflammatory response, and providing neuroprotection [22,23,24]. These mechanisms vary depending on the source of stem cells. The commonly used stem cells are human pluripotent stem cells (hPSCs), fetal-derived neural progenitor stem cells (fNPCs) and mesenchymal stem cells (MSCs).

### 2.2. Human Pluripotent Stem Cells (hPSCs)

Stem cells such as hPSCs can self-renew and differentiate into specialized tissue types making them useful in regenerative medicine. The two most widely used subtypes of hPSCs include human embryonic stem cells (hESCs) and human induced pluripotent stem cells (hiPSCs). hESCs are obtained from the blastocyst of the growing embryo whereas hiPSCs are generated from somatic cells by inducing the expression of four transcription factors: Oct3/4, Sox2, Klf4, and c-Myc [25,26].

The primary use of hPSCs in the clinical setting is through expansion and cell replacement. This cannot occur by direct implantation due to the high risk of cancer by mutations accumulated during the proliferation of undifferentiated tissues as well as the effects of the local microenvironment [27]. Ideally, the cells need to be cultured and differentiated into tissue prior to transplantation. These cells can then be used in various CNS conditions involving the loss of neurons such as Parkinson’s disease, spinal cord injury, and Huntington’s disease [28,29,30]. However, these therapies have a risk for immunogenicity even with the use of autologous stem cells [31].

### 2.3. Fetal-Derived Neural Progenitor Stem Cells

fNPCs are cells that are obtained from the fetal brain and spinal cord [32]. These stem cells possess the ability to differentiate into various CNS cell types such as neurons, glial cells and neuroectodermal cells, making them useful for CNS pathologies [33,34,35]. However, a major limiting factor is the availability of fNPCs; some transplantations may require ten aborted fetuses for one patient [32,36].

The major mechanism of therapy for fNPCs is cell replacement. In primate models of spinal cord injury, intraparenchymal injection of fNPCs proved efficacious and safe with histologic evidence of nerve growth in the injured region [37]. Similar studies have been replicated in rat models of Parkinson’s disease, Huntington disease, traumatic brain injury, and stroke with promising results [38,39,40,41]. These pre-clinical studies on fNPCs have led to clinical trials for a variety of CNS pathologies including amyotrophic lateral sclerosis, traumatic cervical spinal cord injury, and stroke [42,43,44].

### 2.4. Mesenchymal Stem Cells

Mesenchymal stem cells (MSCs) are multipotent self-renewing stem cells that are commonly obtained from the bone marrow but can also be found in the umbilical cord, peripheral blood, and adipose tissue [45]. MSCs have been tested mostly in animal models with pathologies of the heart, liver, eye, and blood [46]. However, there are clinical trials using MSCs for conditions such as amyotrophic lateral sclerosis, stroke, spinal cord injury, multiple sclerosis, traumatic brain injury, and epilepsy [47,48,49,50,51,52]. MSCs delivered intravascularly have a high degree of safety with a meta-analysis showing no increase in acute infusional toxicity, organ system complications, infection, death, or malignancy [53].

## 3. Epilepsy

### 3.1. Overview

Epilepsy is a CNS condition that involves recurrent seizures (≥2) more than 24 h apart, an unprovoked seizure with a probability of subsequent seizures, or a diagnosis of a type of epilepsy syndrome. It affects over 70 million people around the world with the highest risk groups including infants and older aged adults [14]. Causes of epilepsy are broad and up to 60% of patients have an idiopathic cause of epilepsy [54]. Common etiologies include temporal lobe lesions, cerebrovascular disease post-stroke, primary or metastatic brain tumors, vascular malformations, prior CNS infection such as neurocysticercosis, head injury, and Alzheimer’s disease [55].

### 3.2. Mechanism

The pathophysiology of epilepsy can be viewed as a shift in the balance of GABA and glutamate neurotransmission with an increase in glutamate excitatory neurotransmitters due to loss of GABAergic neurons after epileptic insults such as strokes, traumatic brain injury, and status epilepticus. The neuronal circuits are often reorganized to favor abnormal connections -such as in the granule cells of the dentate gyrus collectively called mossy fiber sprouting. The deficit in GABA signaling and enhancement of glutamate signaling is one of the goals in pharmacotherapy [56]. However, research has shown that the mechanism of epilepsy is more complex and may involve several neuropeptides. Neuropeptides are important in the mechanism of epilepsy with their release dependent on neuronal activity, typically released during high neuronal firing frequencies, as opposed to neurotransmitters [57]. In the in-vitro single neuron epilepsy model, the firing rate of neurons is abnormally high compared to normal neurons. As a result, targeting neuropeptide receptors has become an alternative option for future pharmacotherapy [58].

Partial onset seizures are the most common form of adult seizures with temporal lobe epilepsy as the most common subset. Temporal lobe epilepsy is thought to be partially caused by the dysregulation of the amygdala and hippocampus activity. Owing to its structural pattern, the hippocampus is one of the most vulnerable foci in the brain for epileptogenesis [59]. In addition, hippocampus and amygdala play a pivotal role in long-term potentiation and memory consolidation [59]. Therefore, deficits in declarative and spatial memory have been specifically implicated with enhancement of emotional memories when these areas are targeted [59]. Deficits in fear conditioning are also seen in temporal lobe epilepsy in those undergoing unilateral lobectomy of the temporal lobe that includes parts of the amygdala and hippocampus [60]. Neuronal loss in either the hippocampus or amygdala leads to damage in both areas due to the interconnectivity; therefore, when the seizures occur in one region, mirror foci develop contralaterally [61].

Animal studies have reported the posteromedial and posterolateral cortical nuclei disappear in temporal lobe epilepsy with severe neuronal loss in addition to damage to a large projection in the lateral amygdala-hippocampal area that provides emotional sensory input for hippocampal processing [62]. The emotional significance of memories could be affected in temporal lobe epilepsy if hippocampal-dependent. The lateral nucleus used for auditory inputs and fear conditioning was additionally affected, suggesting temporal lope epilepsy can lead to behavioral impairments in fear conditioning. The medial nucleus remains intact and has been proposed to be involved in seizure initiation [63].

Seizures in temporal lobe epilepsy are facilitated via decreased inhibition, increased excitability, and decreased seizure threshold [64]. The decreased seizure threshold is caused by a loss of GABAergic neurons, specifically GABAergic somatostatin containing non-pyramidal neurons. The destruction of the GABAergic somatostatin containing neurons increases the spread of seizures and decreases the synchronized activity of the amygdala-hippocampal neuronal circuitry for emotional declarative memory and fear conditioning [65].

Kindling is a widely used model for describing the development of temporal lobe epilepsy [66]. The model describes how a single seizure can increase the likelihood of subsequent seizures as the seizure threshold lowers. Kindling leads to a lasting change in brain function, predisposing to neuropsychiatric symptoms [67]. A new study in 2021 on mice used optokindling to activate pyramidal cells in the piriform cortex which disrupted GABA production in feedback inhibitory cells, thereby increasing seizure severity and frequency [68]. Alterations in neuronal circuitry after recurrent seizures could be a potential target for therapy to reduce the adverse effects of epilepsy and the progression of kindling, thereby, decreasing neuropsychiatric symptoms [68].

### 3.3. Stem Cell Therapy for Epilepsy

While pharmacotherapy can be used to treat epilepsy, it can induce detrimental side effects and is largely limited to symptomatic treatment rather than prevention. Surgery, while effective in reducing seizure initiation, is invasive with numerous adverse effects and may not be a viable treatment option for all patients. Stem cell therapy has been considered as an alternative to medications and surgery and has proven to be effective in addressing other neurological disorders, such as spinal cord injury and stroke [69]. In a study conducted in Belarus, 22 patients with refractory epilepsy were split into 10 patients in a stem cell therapy group and 12 patients in a control group [52]. A total of 70% of the stem cell therapy group showed transformations of generalized tonic-clonic seizures to simple or complex partial seizures, 50% showed improvement in cognitive status, and 60% demonstrated improvement in anxiety. There was a significantly higher number of responders in the cell therapy group compared to the control. Further, the cell therapy group had a significant reduction in monthly seizure frequency, seizure severity and anxiety compared to control group [52]. In another pilot study, one patient, who initially had epileptic seizures 20–40 times per week, had the episodes decrease in frequency, to 14 episodes per week, after two rounds of bone-marrow derived CD271 + MSCs transplantation [17].

hESCs have the capacity to differentiate into different cell types of the three germ layers and can replace damaged neural cells with healthy stem cells. A study using mice models of status epilepticus showed that neural progenitor cells have the capacity to differentiate into mature neurons after transplantation into the hippocampi [18]. Further research trials suggest hESCs could be used as a means of treatment and prevention of epilepsy with MSCs triggering the release of neurotrophic factors and immunomodulation to reduce the occurrence of seizures [70]. In addition, MSCs have neuroprotective effects by suppressing glutamate toxicity and oxidative injury [70]. An in vitro study demonstrated that MSCs can suppress the expression of the glutamate receptors, including GluR1 AMPA receptors AMPAR subunit as well as the expression of NR1 and NR2A (both NMDAR subunits), yielding reduced glutamate excitotoxicity and reactive oxygen species (ROS) accumulation [71]. hESCs can significantly decrease the frequency and severity of spontaneous seizures and help improve learning and cognitive defects associated with status epilepticus [72].

Stem cell therapy can yield attenuated neuronal death, increased synthesis of neural cells, decreased microglia and astrocyte reactivity, and modulation of neuroinflammation [73]. As astrocyte dysfunction is a common feature in the mechanism of epilepsy, certain stem cells can differentiate into functioning astrocytes [73]. According to another study, MSC derived exosomes attenuated astrocyte dysfunction and neuroinflammation in mice models of status epilepticus, in addition to demonstrating improvement in cognition, learning, and memory [73]. Another approach to stem cell therapy for epilepsy management includes stimulating GABAergic interneurons, as recurrent epilepsy can diminish the prevalence of GABAergic interneurons responsible for inhibitory control in affected brain regions [74]. Cells derived from medial ganglionic eminence (MGE) that are GABAergic progenitors migrate after grafting into the different layers of the hippocampus, leading to a reduction of spontaneous recurrent seizures and reduced abnormal neurogenesis as demonstrated by a mouse model of temporal lobe epilepsy [75]. The formation of synapses by axons of graft-derived GABAergic interneurons with the dendrites of the host CA1 pyramidal neurons plays a significant role in the reduction of spontaneous recurrent seizures, as does reduced EEG amplitude in between seizure episodes [75].

Epileptic seizures can induce anomalous neurogenesis, apoptosis, and neuronal impairment, along with limited GABAergic receptor expression [64]. In epilepsy, the induction of neurogenesis is correlated with significant cognitive impairment, anxiety, and depression [15]. Modifying the expression of neurotrophic factors like brain-derived neurotrophic factor (BDNF) and glial cell line-derived neurotrophic factor (GDNF) could be a potential approach for stem cell therapy [68]. Endothelial progenitor cells are known to increase the expression of BDNF and autophagy-related proteins while improving neuronal circuitry, thereby reducing impairment to learning, memory, and anxiety [16]. Furthermore, in a research study, after one and/or multiple rounds of CD271+ bone marrow MSCs implantation, patients with prior extrapyramidal abnormalities experienced more severe emotional reactions and facial expressions, stronger motor development, better speech, and increased eye and head reactivity to visual and auditory stimuli [17]. Through various target neuronal circuitry such as neurotrophic factors, glutamate receptors, GABAergic interneurons, and astrocytes, stem cell therapy is considered effective in the management of epilepsy. Consequently, it can be utilized to treat neuropsychiatric conditions with similar pathophysiological mechanisms, including but not limited to PTSD.

## 4. PTSD

### 4.1. Overview

PTSD is the persistence of intrusive recollections, avoidance of trauma-related stimuli, negative alterations in mood, and hyperarousal symptoms after exposure to an exceptionally threatening or horrifying event or to prolonged trauma [1]. While many people show capacity to recover after exposure to trauma, it remains a challenge to predict who will develop PTSD [76]. Patients are at increased risk of experiencing poor physical health including somatoform, cardiorespiratory, musculoskeletal, endocrine, gastrointestinal, genitourinary, integumentary, and immunological disorders burden [2]. It is associated with substantial psychiatric comorbidity, increased risk of suicide, and considerable economic burden [2].

### 4.2. Prevalence of PTSD and Treatment-Resistant PTSD

Under DSM-5 criterion A, the prevalence of exposure to potentially traumatic and other life events over the course of a lifetime is estimated at 89% [77,78]. Up to 3% of adults have PTSD at a given moment [79]. Lifetime prevalence rate is between 1.5% and 8.8% and in regions of conflict this rate doubles [3,4,5]. More than half of rape survivors are affected by the condition [80]. Approximately 33% of people with PTSD have treatment-resistant PTSD. Patients undergoing cognitive behavioral therapy may have non-response rates as high as 50%, and for SSRIs about 20–40% [81]. In a study of PTSD patients receiving treatment in a primary care setting, the course of the disorder was chronic, with a recovery rate of 38% and a recurrence rate of 30% [82].

### 4.3. Neuroanatomy and Pathophysiological Mechanism of PTSD

In healthy individuals, stress activates the amygdala, hippocampus, and rostral anterior cingulate cortex for appropriate consolidation of fear memory and extinction of fear. This includes encoding the explicit memory of the stressor, identifying safe contexts relative to the stressor, and habituating appropriately to the stressor (Figure 1). In PTSD, stress hyper-activates the amygdala and there is reduced functional top-down governance of the hippocampus and rostral anterior cingulate cortex over the amygdala, leading to over-consolidation of fear memory and impaired fear extinction.

Abnormal interactions between the amygdala and the hippocampus and mPFC are thought to underlie PTSD, resulting in over-consolidation of fear-based memories and/or weakened fear extinction, which is the decrease in the conditioned fear response [83,84]. The pathogenic mechanism is thought to be a process of fear conditioning marked by amygdala hyperarousal [84] and reduced mPFC activity, resulting in impaired extinction (Figure 2). Reduced hippocampal/parahippocampal activity is also implicated, resulting in over-generalization of fear to non-threatening stimuli and inability to differentiate safe and threatening contexts [84,85].

Conditioning of fear and its extinction in humans is associated with activation of both the amygdala and the hippocampal/parahippocampal cortices [83,84]. This is corroborated by functional imaging studies of PTSD patients showing reductions in hippocampal/parahippocampal activity and hippocampal volumes [86,87,88,89]. Furthermore, hippocampal/amygdala activity was seen correlated with extinction of conditioned fear by context, indicating that amygdalohippocampal interconnectivity is involved [88]. Consequently, patients with PTSD show reduced ability to process contextual surroundings during extinction of fear [90] and decreased renewal of fear when tested for extinction in a different context from the extinction context [91]. Investigations in human subjects concur with findings from rodent studies indicating that the hippocampal/parahippocampal region plays a major role in distinguishing safe versus threatening contexts and can regulate the activity of amygdala “fear neurons”. As patients undergoing exposure therapies for PTSD commonly experience context-dependent relapse of extinguished fear [92], impeding fear renewal via replacement of hippocampal neurons may be a useful therapeutic intervention in this condition.

### 4.4. Shortcomings of Current Treatments

First-line treatment with trauma-focused psychotherapy is preferred to an SSRI in PTSD. An SSRI is a reasonable alternative for patients who prefer it and when cognitive-behavioral therapy cannot be obtained. Multiple clinical trials have shown that the psychotherapies most effective for this condition include exposure therapy, a combined exposure and cognitive therapy known as trauma-focused cognitive-behavioral therapy, as well as eye movement desensitization and reprocessing [6,93]. However, these therapies have limited availability and inconsistent treatment outcomes. While some patients have a robust response to treatment, others have poor or partial relief of symptoms requiring restructure of regimen based on predominant symptom clusters, treatment availability and/or patient preference. Studies have shown between 18% and 50% of patients with PTSD have a stable recovery within three to seven years, the rest have either a more persistent or chronic course [94]. Furthermore, while it is thought that early treatment of PTSD may prevent its chronicity, this has not been shown empirically, especially for pharmacotherapy [95].

Unfortunately, SSRI may have adverse effects including but not limited to sexual dysfunction, drowsiness, weight gain, insomnia, anxiety, dizziness, headache, and dry mouth [96]. Serotonin norepinephrine reuptake inhibitors (SNRIs) may additionally cause increases in blood pressure, nausea, and diaphoresis among others. Patients may fail to respond to up to two SSRI/SNRI trials until fourth-line treatment with a second-generation antipsychotic (SGA), such as risperidone or quetiapine is indicated. These SGAs have adverse effects including extrapyramidal symptoms, sedation, weight gain, glucose abnormalities, hyperlipidemia, orthostatic hypotension, QTc prolongation, and anticholinergic symptoms as well as others [8,97].

### 4.5. Similarities of PTSD to Epilepsy

In epilepsy, the loss of inhibitory projection neurons leads to alteration in the synchronized oscillatory activity of amygdalohippocampal circuits and further spread of seizure activity via excitation of the basolateral nucleus of the amygdala, ultimately causing dysfunctionality of both fear conditioning and emotional enhancement of declarative memory. Trauma in PTSD causes amygdala hyperarousal and associated decreases in hippocampal and mPFC activity which ultimately produce the nearly opposing effect of over-consolidation of fear-based memories and impaired fear extinction. As it is either the loss or over-functionality of nuclei or their projections that are implicated in both syndromes, it is plausible that use of stem cell therapies modulating the connectivity of this circuit could be of therapeutic benefit in both conditions [98].

Additionally, seizures are precipitated by stressful events or durations in about half of individuals with epilepsy. It is thought that stress hormones play a role in neuronal excitability. Case studies of PTSD patients have shown a pattern of video electroencephalogram recorded seizure activity related in time to stressful occurrences. Thus, stress exposure has been linked to the development of epilepsy and the susceptibility to stress as a seizure trigger. These findings together suggest that stress-induced epilepsy might be more likely to result in stress-precipitated seizures [99]. Furthermore, PTSD is a known risk factor for psychogenic nonepileptic seizure (PNES), and up to one-third of epilepsy patients treated at tertiary care epilepsy centers have PNES as well. Lastly, temporal lobe epilepsy (TLE) is commonly misdiagnosed as PTSD and vice-versa [99].

### 4.6. Behavioral Sensitization and Electrophysiological Kindling

The process of behavioral sensitization is thought to occur when emotional trauma is connected with episodes of depression, leaving traces of electrophysiological “kindling” which persist after remission. The scar theory of depression presumes that even under moderate or no psychosocial stress, this “scar” of negative concept neuronal connectivity might increase susceptibility to the onset of new depressive episodes [100]. This is supported by associative network theory which predicts that as connections between negative concepts strengthen the “scarring” increases, ultimately lowering threshold for activation and thus increasing susceptibility [100]. These findings together suggest that traumatic experiences and associated strengthening of connections relating to negative concepts could lead to neuronal hyperresponsiveness precipitating seizure-like activity. This relative excitation could potentially be a target of stem cell therapy, as neuronal regeneration could weaken those connections made by trauma just as it does for epileptogenic foci in epilepsy.

### 4.7. Stem Cell Therapy in Memory Symptoms, Neuropsychiatric Disorders, and PTSD

Stem cell therapy for PTSD has been studied in the brains of rats in which human iPSCs were differentiated into neural progenitor cells (NPCs). Induced pluripotent stem cell-derived neural progenitor cell transplantation was shown to promote regeneration and functional recovery after PTSD in rats [101]. It is thought that PTSD affects the brain at a cellular level by reducing the number of GFAP immunoreactive cells as well as decreasing production of BDNF, resulting in hippocampal neuron injury. Hippocampal astrocytes in rats with PTSD manifest morphological alterations including a change in cell polarity towards a more fusiform shape [102]. The animal model showed efficient differentiation of iPSCs into NPCs and glial cells in vivo to replace damaged hippocampal neurons (Table 1). The transplanted cells enhanced the expression of mature neurons, as measured by NeuN levels. These cells also over-expressed BDNF and GDNF, neurotrophic and neuroprotective factors that suggest the potential for further neurogenesis capability (Figure 3).

BDNF has been shown to be significantly decreased in animals following PTSD [103]. Memory deficits as a result of PTSD cause reduction in BDNF expression in the hippocampus and deficient performance in hippocampus-dependent tasks. This suggests that an increase in BDNF expression mediates the beneficial effects of iPSC-NPC transplantation. The cells also expressed GFAP, a cytoskeletal protein thought to be necessary for recovery processes following injury to the brain. This occurs through the remodeling of astrocytes in response to different physiological and pathological situations [104].

Induced pluripotent stem cell-derived astrocytes were shown to generate increased downstream cytokine production (including IL-1 and TNF) when exposed to IL-1B [105]. They have potential to be utilized in management of PTSD, as immune system dysfunction is a common comorbidity, in particular elevated levels of norepinephrine and impaired glucocorticoid receptor signaling [106]. Modulation of promoters in the glucocorticoid receptor gene can influence gene expression and receptor affinity, thereby serving as a potential mechanism for stem cell modification in the treatment of PTSD. Furthermore, the hypothalamic-pituitary axis tends to be upregulated with sustained increases in corticotrophin-releasing factor in PTSD, and could serve as a target for stem cell therapy [106].

Stem cell therapy has additionally been implicated for treatment in models of neuropsychiatric disorders; however, studies applying stem cells to these disorders are scarce. A study using stem cell therapy in mice found improvement in memory function in mice with brain damage from replicated human diseases such as strokes and dementia. The study used glial cells to stimulate repair mechanisms within the brain to support neurons and limit progression of brain damage. It showed enhancement in new neural connections and myelin production to protect the connections. The study was able to reprogram iPSCs back to an embryonic stem cell-like state. When the cells were removed from the mice after a few months, the repair in the brain was not reversible, suggesting that there is not a reliance on the transplanted stem cells long-term for the treatment to be effective [107].

In Alzheimer’s disease, neural precursor cells have been implicated in mice for behavior restoration. These stem cells have been linked to improvements in memory, learning, and behavioral dysfunction [108]. Umbilical cord blood CD34+ cells stimulate angiogenesis leading to positive outcomes in cerebral ischemia mice models, improving memory deficits [109]. BDNF has also been shown to increase the effects of neural precursor cells in rat brains, leading to cognitive improvement related to increased hippocampal synaptic density [110]. In rats, mesenchymal stem cells were injected in the stroke-damaged areas, leading to improved neurobehavioral function and reduction in the volume of the stroke lesion [111]. Inflammatory mononuclear phagocyte microglia have also been studied in their production of neurodegenerative proinflammatory cytokines in Alzheimer’s disease [110]. Some studies, on the other hand, have found neuroprotective functions of the microglia such as eliminating plaques and releasing neurotrophic agents [112]. In mice, applying mesenchymal stem cell improved cognitive function and microglia activation, decreasing the inflammatory response [108].

Clinical cases of autistic children receiving CD34+ cells have shown positive responses in areas of the brain undergoing hypoperfusion and ischemia related to the neurophysiology of autism [113]. Additionally, the abnormal migration and reduction in GABAergic interneurons during prenatal development leads to neuronal dyssynchrony in autism, epilepsy, and schizophrenia. In studies using mouse xenografts, human stem cell-derived interneuron precursors were able to differentiate in vivo [114]. When transplanted in rat and mouse models for epilepsy, hPSC-derived interneuron precursors fired action potentials and developed synaptic connections that reduced abnormal seizure activity. However, minimal migration was observed after four to seven months post-transplantation in the host mouse brain. The main benefits of interneuron grafts in these models were seen after numerous months post-transplantation [114]. Human PVECs were also implanted into the host mouse brains, mimicking mouse angiogenesis and allowed for migration of human GABAergic interneurons after transplantation. This allowed for improvement in cell migration and reduced GABA levels that were being released by the brain [115]. This further led to an improvement in the behavioral outcome in preclinical psychiatric disorder models within a month of transplantation, compared to the several months observed with solely interneuron precursors [116]. With cotransplantation, the prenatal forebrain developed angiogenesis and showed potential for regeneration and repair in adult brains to help improve psychiatric behaviors, particularly behavioral function [117].

## 5. Conclusions: Stem Cell Therapy for PTSD

Stem cells are currently being employed in the treatment of focal epilepsies that share similar neuroanatomical targets with PTSD. Targeting the amygdala in epilepsy using stem cells provides benefit by reducing the complications, centers of epileptic activity and neuropsychiatric comorbidities of epilepsy. It follows that such therapy could be used in conditions which have a similar neuronal basis, such as PTSD. Animal models in rats have provided proof-of-concept for how stem cells can influence functional outcomes in PTSD, as transplanted induced-pluripotent stem cells have shown the capability to differentiate into and replace damaged neuronal and glial tissue, secrete protective neurotrophic factors, and promote neuronal regeneration. Additionally, stem cells could help re-establish regulation of the abnormal pro-inflammatory cascade implicated in PTSD and potentially mitigate development of the condition in addition to reducing symptom intensity. Stem cell therapy could potentially be used for treatment resistant PTSD with milder side effects compared to current treatment regimens. Now that animal research has begun to demonstrate the utility of stem cells in treating PTSD, it has opened doors for future clinical trials.

## Figures and Tables

**Figure 1 diseases-09-00077-f001:**
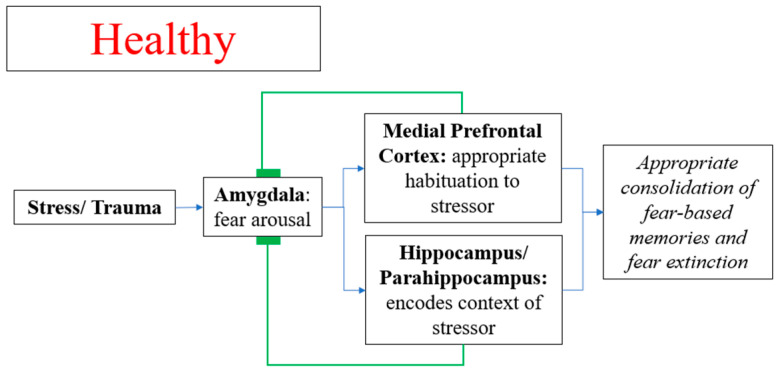
Consolidation of fear memory and extinction of fear in healthy individuals.

**Figure 2 diseases-09-00077-f002:**
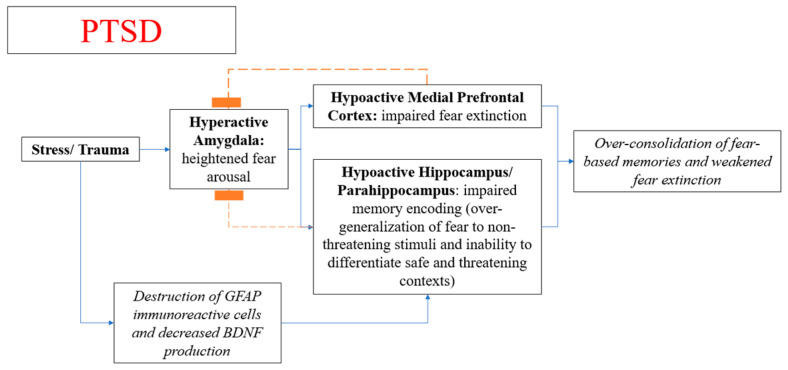
Abnormalities in consolidation of fear memory and extinction of fear memory in individuals with PTSD.

**Figure 3 diseases-09-00077-f003:**
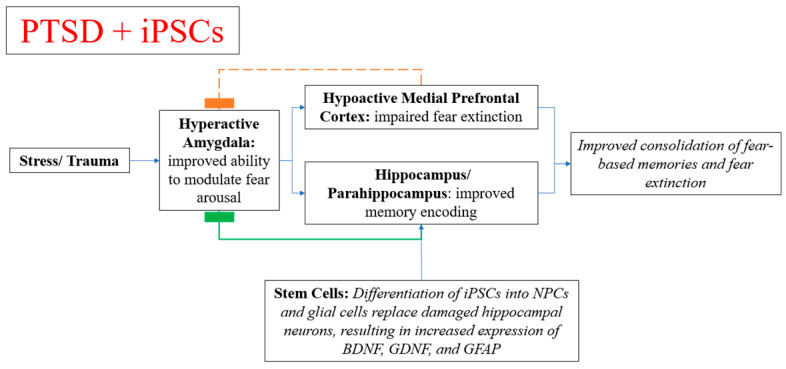
Mechanism of potential beneficial effects from stem cell therapy in individuals with PTSD.

**Table 1 diseases-09-00077-t001:** Summary of results from Liu et al. [101]. Legend: PTSD—post traumatic stress disorder group (as induced by previously established protocol), PBS—phosphate buffer solution (does not contain any stem cells), iPSC—induced pluripotent stem cells, Day #—indicates number of days post-transplantation.

Variables Tested	Control	PTSD	PTSD + PBS	PTSD + iPSC Day 7	PTSD + iPSC Day 14	PTSD + iPSC Day 21
Total distance in Open Field Test	No change	Significantly decreased relative to control	Significantly decreased relative to control	Increased relative to PTSD	Increased relative to PTSD	Increased relative to PTSD
Interest area stay time in Open Field Test	No change	Significantly decreased relative to control	Significantly decreased relative to control	No significant change from PTSD	Significantly increased relative to PTSD	Significantly increased relative to PTSD
Behavior modification in Open Field Test	No change	Significantly decreased relative to control	Significantly decreased relative to control	No significant change from PTSD	Significantly increased relative to PTSD	Significantly increased relative to PTSD
Freezing time in Fear Conditioning Test	No change	Significantly increased relative to control	Significantly increased relative to control	Significantly decreased relative to PTSD	Significantly decreased relative to PTSD	Significantly decreased relative to PTSD
Astrogliosis	No change			Increased GFAP (+) astrocytes compared to control		Significantly increased GFAP (+) astrocytes compared to control and PBS
NeuN neuronal maturation marker expression	No change	Significantly decreased expression relative to control	Significantly decreased expression relative to control	Increased relative to PTSD+PBS	Increased relative to PTSD+PBS	Increased relative to PTSD+PBS
BDNF expression	No change	Significantly decreased expression relative to control	Significantly decreased expression relative to control	Slightly increased expression relative to PTSD	Significantly increased expression relative to PTSD	Significantly increased expression relative to PTSD

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
