# Peer review of "Stem Cell Therapy for Post-Traumatic Stress Disorder: A Novel Therapeutic Approach"

_diseases, 2021, doi:10.3390/diseases9040077_

Round 1

Reviewer 1 Report

This is a good review. I enjoyed reading it.

Reviewer 2 Report

The manuscript entitled "Stem Cell Therapy for Post-Traumatic Stress Disorder: A Novel Therapeutic Approach" by Gala et al. It will be improved if the followings are undertaken.

- The introduction part can be elaborated.

- It will be interesting if a timeline was drawn to show the past and present progress of regimen on treating PTSD.

- The manuscript would be benefited if tables are designed to summarize some key and interesting data/numbers, such as cell/animal model used as well as year of study, and (pre)clinical trials details. In addition, it will be nice if a table was designed to let reader compare the commons and differences between PTSD to epilepsy/alzheimer/other neuronal conditions.

- The manuscript would be benefited if figures are designed to let readers visualize your ideas than just the plain text.

- References should be recheck for accuracy as some references are displayed inappropriately, such as reference number 63, 66, 101, 102, and 105, where the terms "!!! INVALID CITATION !!!" were shown.

- Typos and unfriendly mode of English can be found. The authors are suggested to conduct proofreading of the manuscript by native English speaker(s) before resubmitting the revised manuscript.

Reviewer 3 Report

Dear Authors 

This paper is well structured but looks quite boring without any illustration for explaining the core of the topic theme. Also, in the text, you can share the previously reported work images and cases by taking permission from authors.

Conclusion is not well written improve it. 

Also, the reference style is not according to the journal guidelines.  

Reviewer 4 Report

The manuscript “Stem Cell Therapy for Post-Traumatic Stress Disorder: A Novel Therapeutic Approach” by D. Gala and co-authors is a well-written and detailed overview of current advances in stem cell therapy in epilepsy, memory symptoms, neuropsychiatric disorders, and post-traumatic stress disorder (PTSD). The authors focused on the rationale for the use of stem cells for the treatment of PTSD. The topic of the review is highly relevant. The manuscript is very interesting, written at a high level, easy to read and understand. The review covers a large amount of literature data and makes a significant contribution to the systematization of knowledge. The authors cited a large number of research articles, a significant portion of which have been published over the last five years. The only drawback of the manuscript is the lack of summarizing tables or illustrations. To improve the perception of information by the reader, it would be useful to add to the manuscript a schematic illustration of the mechanisms of pathogenesis of PTSD or a table summarizing the positive effects of stem cell therapy for PTSD in animal models (with appropriate references).

Round 2

Reviewer 2 Report

  • Some of my previous comments are still not addressed.
  • Typos and unfriendly more of English still can be found, the authors are advised to proofread the whole article by native English speakers.
  • Some of the references in the references section are misaligned.

This manuscript is a resubmission of an earlier submission. The following is a list of the peer review reports and author responses from that submission.

Round 1

Reviewer 1 Report

The paper is original and well-written. English style is good. Methods and statistical analysis is reproducible. Results are relevant. References are good.

Reviewer 2 Report

The authors discussed the stem cell therapy for epilepsy and PTSD with the idea that these diseases have similar pathophysiology.

Unfortunately, I cannot agree with this underlying idea at all and the reference which the authors referred to is not appropriate for this idea (I mean reference 2). Therefore, I cannot recommend this review article for publication.

Reviewer 3 Report

This is an interesting idea for a review and it is generally well written. There are some minor issues that are detailed below, but towards the end of the article I felt that the point the authors are making is not adequately argued. Please see the below comments for details:

First paragraph, introduction – could benefit from a little bit more information on PTSD – e.g. symptom profile, societal burden, current treatment success rates (etc) that would justify the need to do something extra to treat the disorder

Page 2 lines 79-86: two sentences here are repeated. Is the last sentence of paragraph 1 of Section 2.3 meant to be the last sentence of that entire section?

Page 3 line 98 – fullstop missing after “and blood [36]”

Page 3 lines 100-105, similar to page 2 these sentences are again repeated. I am wondering whether there was additional information in these paragraphs that is missing in addition to copy and paste errors?

Page 4 line 151 – “behavioral impairments in fear conditioning” is missing a fullstop at the end of the sentence

Page 4 line 174 – in the sentence “an alternative to medication and surgery and has proven to be effective” – is “and surgery” meant to be in this sentence, since the topic of the paragraph had already changed to stem cell therapy?

Page 4, line 175 – was the reduction in symptoms in the stem cell group significantly different from changes in the control group (for ref 42)?

Page 4 line 180 – ref 7 is not a clinical trial, either a case study or a pilot study

Page 4 line 187 -  “transplantation into the hippocampi .[8].” – remove fullstop before the reference

Page 4 line 195 – ref 61 is not a clinical trial, and ref 54 (line 201) is also not the right reference there. Strictly speaking ref 64 is not a clinical trial either.

Page 5 last sentence section 4.1 – I am not sure that this sentence is the most important aspect of PTSD that needs to be included in the context that you are describing.

Page 6 – “Various authors have suggested that in the general population about 33% of people have treatment-resistant PTSD.” – I don’t think that this is accurate and there is no reference for this provided. If only 3% of adults have PTSD then 33% of the general population couldn’t have treatment resistant PTSD – or do you mean that 33% of the PTSD population?

Page 6 line 261 – the term fear extinction is not described and a general audience may not be familiar with this term – suggest defining in this paragraph briefly

Section 4.4. – SSRIs are abbreviated in the first paragraph as SRIs and then in the next paragraph as SSRIs. Also in this and the preceding section I think the authors need to better justify why SSRIs are not effective in PTSD, make sure to cite the right papers (e.g. I think there are a few reviews on the topic that show overall low-moderate efficacy, and also discuss the idea that SSRIs don’t target the memory mechanisms such as fear extinction during recovery)

My most major comment would be this: After reading the full review, I am not as convinced as I had hoped when reading the abstract that there is convincing evidence that stem cell therapy might be useful in PTSD. Firstly, there is not sufficient elaboration of the neural pathways involved in both disorders, and to my knowledge the neural mechanisms are much more different between the disorders than the authors imply. Specifically, seizures in epilepsy are the result of electrical abnormalities in the brain whereas the structural changes in PTSD do not produce the same kind of phenomenon – in fact, even in PNES there is no observed electrical properties of the seizures. That being said, I am not unconvinced that the idea is misled. I would suggest the authors need to argue the point more, providing more information on things such as: (a) Are there any studies at all looking at memory symptoms, PTSD, or other psychiatric conditions in animals that use stem cell therapy? (b) Going into more depth on the neural mechanisms of PTSD would be helpful, see reviews such as Fullana et al 2018, Milad & Quirk 2012, etc that cover the topic more comprehensively; (c) covering the biological mechanisms proposed to be involved in PTSD fear extinction memory, e.g. there is a large literature on BDNF in PTSD and the potential mechanisms are well described elsewhere (starting point perhaps Pitman et al 2012) – by focusing less on the structural characteristics of brain abnormalities in PTSD and more on the synaptic mechanisms you might be able to develop a stronger argument at the end of the article. As it is, the review is largely focused on epilepsy and stem cell therapy – the PTSD section at the end is really quite brief considering the topic of the article. Also, the authors miss some key articles, e.g. El-Jawahri et al 2015 and articles summarised in that article (and perhaps some that cite that article too)

Round 2

Reviewer 2 Report

I also cannot agree with “Similar neuroanatomic locations as well as neural networks” at all. In my opinion, PTSD and epilepsy is very different. Also, the authors show no literature support to their idea.

Therefore, I cannot recommend this article for publication.

Reviewer 3 Report

No further comments, I am happy with the revisions. I think that this is a very interesting idea and will be curious to see the results of initial studies.